# Unveiling the mechanism of triphos-Ru catalysed C−O bond disconnections in polymers

Alexander Ahrens [1] ✉, Gabriel Martins Ferreira Batista [1],
Hans Christian D. Hammershøj[1], Emil Vincent Schwibinger [1],
Ainara Nova [2] ✉ & Troels Skrydstrup [1] ✉

Ruthenium complexes with facially coordinating tripodal phosphine ligands are privileged catalysts for a broad range of (de-)hydrogenation-based transformations. Among these, C−O bond hydrogenolysis holds potential for the depolymerisation of both the biopolymer lignin and epoxy resins applied in wind turbine blades, aircrafts and more. However, this methodology is poorly understood in mechanistic terms. Here, we present a detailed investigation on the triphos-Ru catalysed C−O bond scission on a molecular level. A combination of experimental, spectroscopical and theoretical studies elucidates the reactivity of the ruthenium trimethylenemethane precatalyst, revealing the key roles of ruthenium phenolates in both catalyst activation as well as the catalytic cycle itself. Furthermore, a Ru(0)/Ru(II) oxidative addition into the C−O bond is disclosed, with a triphos-Ru(0) dihydrogen complex as entry point. With the molecular nature of the operating triphos-Ru species and the thermodynamics and kinetics of the catalysis unravelled, improvements of established methods as well as design of related transformations may become possible.

Using biomass as feedstock for chemical products as well as efficient recycling strategies for plastics have been identified as key factors in establishing a sustainable chemical industry[1–3]. The depolymerisation of lignin has gained much attention, not only as it represents an abundant byproduct of the pulp and paper industry, but also because it contains valuable aromatic scaffolds[4]. Depolymerisation strategies for plastic-based materials are an important approach for reducing the consumption of resources and the accumulation of waste[1,3]. Recovering polymer building blocks from end-of-life plastics via selective bond disconnections would allow for their reintroduction into existing production chains, yielding virgin-grade polymers[5–7]. This is especially interesting for thermoset plastics such as epoxy polymers, *e.g.* used for the construction of wind turbine blades and airplanes, as no alternative recycling methods are viable for these structures[8,9].

The triphos-Ru catalysed hydrogenolysis of C−O bonds has been demonstrated for both, disassembling lignin models[10], and the selective depolymerisation of commercial epoxy resins and their composites[11] (triphos = 1, 1, 1-tris(diphenyl-phosphinomethyl)ethane). For lignin model compounds, Bergman, Ellman et al. initially reported the activity of a xantphos-Ru catalyst[12]. Triphos-Ru-TMM complexes (TMM = trimethylenemethane) were then found to be highly efficient (pre)catalysts for disconnecting the C(alkyl)−O bonds in β-O-4 linkages and recovering phenols (Fig. 1a)[10]. On more elaborate lignin model compounds, a competing C−C bond scission was observed for triphos-Ru catalysts[13], while the xantphos-based system proved susceptible to deactivation[14]. Recently, our group demonstrated a recycling concept for epoxy resins and their composites, also relying on triphos-Ru-TMM as precatalyst (Fig. 1)[11]. By selectively disconnecting C(alkyl)−O bonds

[1]Department of Chemistry and Interdisciplinary Nanoscience Center (iNANO), Aarhus University, Aarhus C, Denmark. [2]Department of Chemistry, Hylleraas Centre for Quantum Molecular Sciences and Centre for Materials Science and Nanotechnology, University of Oslo, Oslo, Norway.
✉e-mail: aahrens@inano.au.dk; a.n.flores@kjemi.uio.no; ts@chem.au.dk

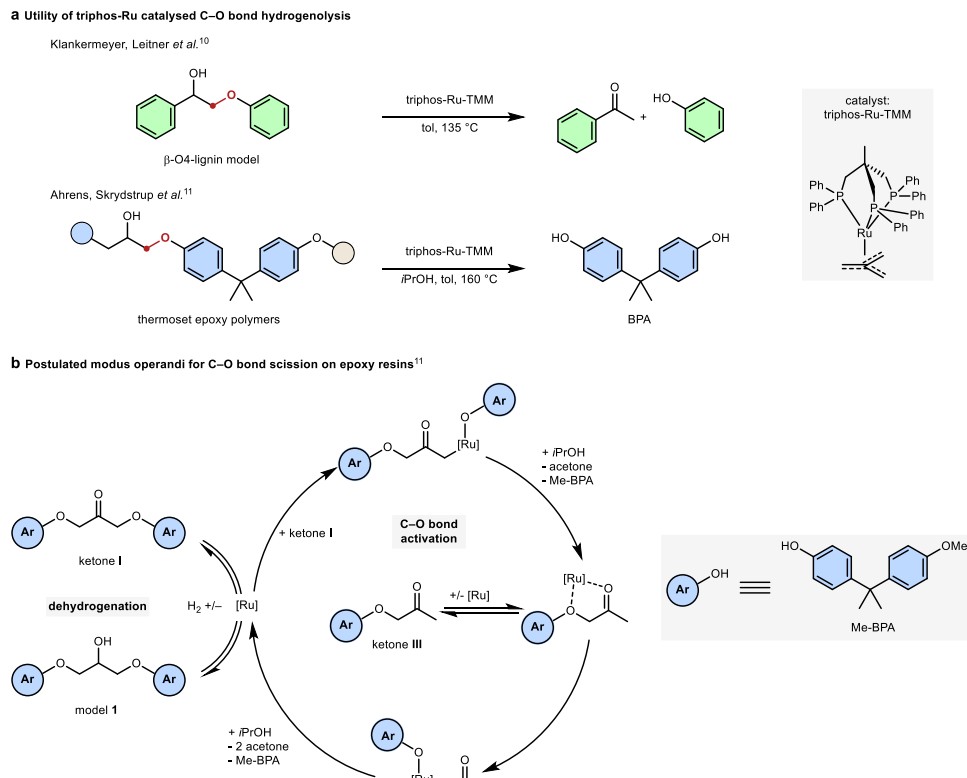

**Fig. 1 | Reports on triphos-Ru catalysed C−O bond scission. a** Disconnection of C(alkyl)−O bonds (marked red) in lignin models and epoxy resins, respectively. **b** Postulated mode of reaction for the deconstruction of epoxy model **1**. BPA bisphenol A, TMM trimethylenemethane, triphos 1,1,1-tris(diphenyl-phosphino-methyl)ethane, Me-BPA O-methyl bisphenol A.

in the polymer matrix, one of the major polymer building blocks, bisphenol A (BPA), along with glass or carbon fibres can be recovered in high quality from end-of-life structures, such as state-of-the-art wind turbine blades.

Generally, the triphos-Ru fragment has been demonstrated to be a privileged motif, catalysing a multitude of transformations relevant to sustainability. Among these are hydrogenations of polyesters/polycarbonates[15] and carbon dioxide[16–18], the reforming of biomass-derived chemicals[19,20], as well as aminations of alcohols[21,22]. For example, in the direct hydrogenation of carbon dioxide to methanol, a detailed understanding of the reactivity of the catalytic intermediates helped greatly improve the reactivity by rational catalyst design[23,24]. However, on both lignin models and epoxy polymers, a mechanistic understanding of the Ru-catalysed C(alkyl)−O disconnection is lacking[10,11], thus hindering the advancement of these important concepts. The postulated mode of reactivity entails an initial dehydrogenation of the central alcohol group with the conversion of model **1** to ketone **I** (Fig. 1b), resulting in the weakening of the adjacent C−O σ-bond. Analysis of bond dissociation energies and experimental studies support the viability of this assumption[10–12]. That a low-valent Ru species subsequently inserts into the weakened C−O bond (Fig. 1b) has been speculated on[11,12,14]. However, oxidative additions into C−O bonds by a Ru(0) complex are rare and have, to the best of our knowledge, not been mechanistically investigated. Furthermore, for protocols applying triphos-Ru-TMM, the catalyst activation as well as the nature of the active species have not been described[10,11]. A lack of understanding regarding the elemental steps of the catalytic cycle, the catalyst activation, as well as the nature of the active ruthenium species obstructs improving the reported methodologies or envisioning new transformations by rational design. In this report, we set out to reveal the details of the catalytic cycle on a molecular level using both experimental and theoretical

methods. Our studies disclose the central role of triphos ruthenium phenolate and hydride complexes for catalyst activation as well as the catalytic cycle. Furthermore, DFT (density functional theory) studies confirm a kinetically and thermodynamically viable Ru(0)/Ru(II) mechanism for the C−O bond scission via oxidative addition.

## Results

### Reactivity of the TMM-precatalyst

During our initial investigations on the Ru-catalysis with epoxy models such as model **1** (Fig. 2), we observed the effective C−O bond cleavage leading to a quantitative yield of O-methyl bisphenol A (Me-BPA) in the presence of isopropanol[11]. In the absence of isopropanol, no conversion was observed. If the secondary alcohol acted solely as a hydrogen donor for the reduction of the C−O bond on the intermediate ketone **III** (Fig. 1b), half of the cleavable bonds should have been disconnected instead of the observed lack of conversion. Interestingly, isopropanol or other alcohols are not required to disconnect the C−O bonds in lignin models[10]. In our hands too, a lignin model cleanly disassembled in the absence of isopropanol (Fig. 2a). Operando monitoring experiments using [1]H and [31]P NMR spectroscopy revealed that the lack of conversion on epoxy model **1** without isopropanol was caused by the absence of catalyst activation. No consumption of the precatalyst was observed, while experiments that show the desired C−O bond scission consumed the precatalyst under formation of new species observable in [31]P NMR spectra[11]. In order to investigate the interplay between the alcohol and the precatalyst, triphos-Ru-TMM was suspended in toluene-$d_8$ with 10 equiv of isopropanol followed by heating to 160 °C overnight. No consumption of the TMM complex was observable. Nonetheless, the formation of acetone demonstrates that triphos-Ru-TMM can dehydrogenate secondary alcohols without a preceding activation.

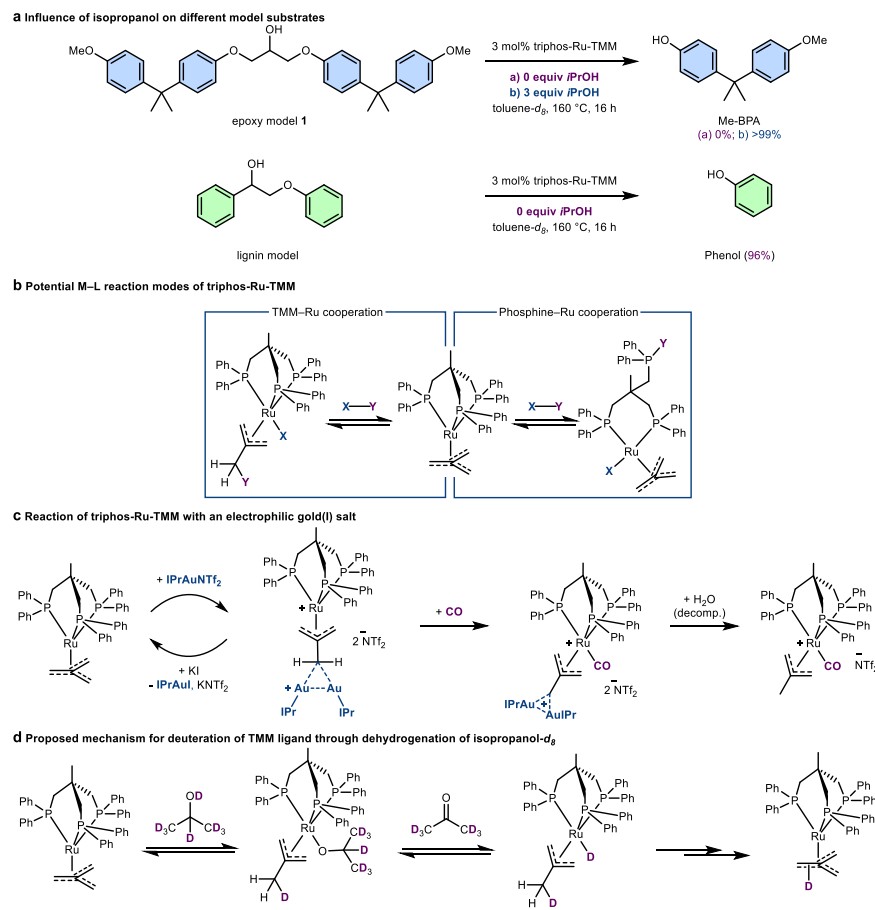

**Fig. 2 | Catalytic disconnection on model compounds and reactivity of TMM ligand. a** Comparison between epoxy model and lignin model compounds. Yields determined using ¹H NMR spectroscopy using 1,3,5-trimethoxybenzene as internal standard. **b** TMM versus phosphine as active ligand. **c** Complexation of TMM ligand by IPrAu cations forming a diaurated 2-methyleneallyl ruthenium complex (proposed structure). Molecular structure of triphos-Ru-methylallyl-CO in the crystal (CCDC 2303540). **d** Proposed mechanism for deuterium incorporation. Ratio of incorporation determined using ¹H and ²H NMR spectroscopy. TMM trimethylenemethane, triphos 1,1,1-tris(diphenyl-phosphinomethyl)ethane, Me-BPA O-methyl bisphenol A.

The reversible protonation of the TMM ligand on ruthenium has been reported using strong acids[25]. Based on this, a metal-ligand cooperation can be postulated with TMM acting as a proton acceptor, forming a 2-methylallyl ligand (Fig. 2b). An alternative metal-ligand cooperation mode of triphos-Ru-TMM could involve the dissociation of one of the phosphine arms. Such arm off/arm on mechanisms have been observed for octahedral iridium (III) complexes with analogous d⁶ configuration[26] and discussed for related triphos-Ru complexes[27]. In order to probe for this reaction modus, triphos-Ru-TMM was left to react under 1 atm of CO at 160 °C overnight. However, no new species, e.g. with CO blocking a coordination site, could be observed. Lastly, it was tested whether switching from a hard proton to a soft electrophile could trap a dissociated phosphine by reacting triphos-Ru-TMM with IPrAuNTf₂ (IPr = 1,3-Bis-(2,6-diisopropylphenyl)imidazol-2-ylidene) (Fig. 2c). With 2 equiv of the gold salt in THF-d₈, the clean transformation into a new complex was observed within minutes. This new species shows a singlet phosphine peak at 48.4 ppm in the ³¹P NMR spectrum, and the signals of the triphos ligand in ¹H NMR spectra correspond to two chemically equivalent IPr fragments.

Instead of forming a gold–phosphine complex, the Lewis acidic gold centres interact with the TMM ligand analogously to a proton (Fig. 2c). Diaurated carbon centres are known to be stable due to favoured aurophilic interactions on gold (I) complexes[28]. Reacting this new species with KI led to the quantitative regeneration of triphos-Ru-TMM, which implies that the Au₂–C bond is dynamic. As the free coordination site on the cationic ruthenium centre would favour the reversibility, IPrAuNTf₂ and triphos-Ru-TMM were reacted under 1 atm

of CO. The ruthenium 2-methylallyl species is trapped by the strongly binding carbonyl ligand, which blocks the reversion to TMM. Here, single crystals suitable for X-ray analysis of the decomposition product could be grown, confirming the structure of a ruthenium 2-methylallyl complex. However, the gold-carbon bond cleaved presumably through formation of gold nanoparticles. The 2-diaurated-2-methyleneallyl ruthenium complex presents a unique case of rare gold allyl complexes[29–32]. Complexes where the TMM ligand bridges metal centres such as ruthenium[33], rhodium[34] or samarium[35] are equally rare. Finally, isotope labelling experiments were conducted. Reacting triphos-Ru-TMM under transfer hydrogenation conditions with a 1:1 mixture of isopropanol-d₈ and acetone-d₆ in benzene under reflux over-night resulted in the deuteration (85% D incorporation) of the TMM ligand. This hints at a potential metal-ligand cooperation mechanism on the Ru-TMM fragment, as isopropanol-d₈ would protonate the ligand reversible before being dehydrogenated (Fig. 2d).

The investigation on the precatalyst's reactivity and the criteria for its activation enabling C−O bond scission was continued on model compounds. When ketone **III**, representing the proposed intermediate of the deconstruction process, was subjected to the reaction conditions in the presence of isopropanol, no formation of Me-BPA was observed. Instead, the starting material had been reduced to alcohol **III** (Fig. 3a). Here too, spectroscopy revealed the absence of catalyst activation, despite the transfer-hydrogenation occurring. This is surprising, as conversion from ketone intermediates was noted in analogous experiments on lignin models[10,12]. From these experiments, we speculated whether the phenol liberated in the first bond scission

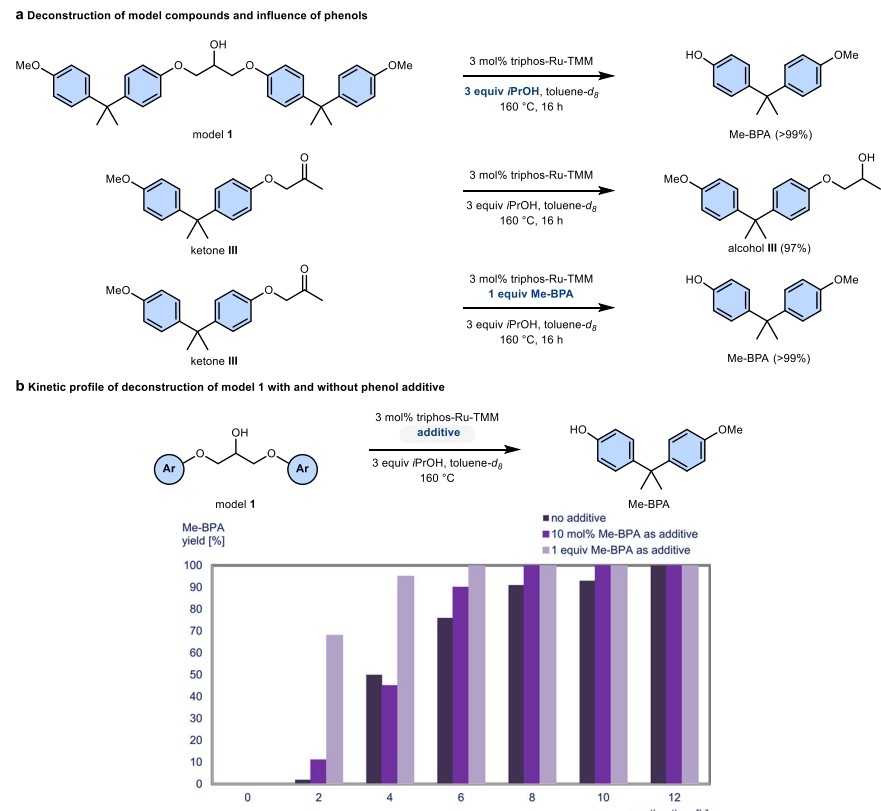

**Fig. 3 | Disconnection on different model compounds and influence of phenol additives. a** Comparison between model **1** and ketone **III**. **b** Kinetic profile of the deconstruction of model **1** with and without Me-BPA as additive. indigo (dark) – no additive, *violet* – 10 mol% Me-BPA, lavender (pale) – 1 equiv Me-BPA. Yields determined using $^1$H NMR spectroscopy using 1,3,5-trimethoxybenzene as internal standard. TMM trimethylenemethane, triphos 1,1,1-tris(diphenyl-phosphino-methyl)ethane, Me-BPA O-methyl bisphenol A.

could play a specific role for C−O bond activation in ketone **III**. Indeed, with the addition of 1 equiv of Me-BPA to the catalytic reaction conditions, clean deconstruction of ketone **III** was observed over a reaction period of 16 h (Fig. 3a).

With the preliminary observations that the presence of isopropanol is crucial for precatalyst activation for C−O bond cleavage in model **1**, and the role of Me-BPA for the σ-bond breakage in ketone **III**, we next turned to in operando experiments applying $^1$H and $^{31}$P NMR spectroscopy in J Young NMR tubes to provide further information for these individual steps. For the catalytic C−O bond disconnection, triphos-Ru-TMM acts as a precatalyst as earlier reported, forming several new species over an induction period of 2 to 4 h on model **1**[11]. After confirming that isopropanol does not transform the precatalyst to the active catalyst, triphos-Ru-TMM was reacted with 4 equiv of Me-BPA in toluene-$d_8$ at 160 °C. After 16 h of reaction time, a significant amount of starting material remained, but several new signals formed according to $^{31}$P NMR spectroscopic analysis of the reaction mixture. Extended reaction times did not lead to further conversion, however, a large excess (12 equiv.) of Me-BPA promoted the formation of a main species with a singlet at 49.6 ppm. Unfortunately, it was not possible to isolate and characterise the species from the mixture. As phenols clearly interact with the precatalyst, we considered whether the addition of an appropriate phenol could reduce the induction time necessary for the activation of the precatalyst. The kinetic profile for the deconstruction of model **1** without additives was compared to the deconstruction using 10 mol% of Me-BPA and 1 equiv of Me-BPA, respectively, as additives (Fig. 3b). In absence of additive (indigo / dark), the catalytic deconstruction shows a 2–4 h induction period, and full conversion after 12 h. Although the reaction rate is marginally

faster using 10 mol% phenol additive (violet), the overall kinetic profile remains similar. However, full conversion is observed after 8 h. In the presence of 1 equiv of Me-BPA (lavender / pale), reaction rates increase sharply. After 2 h, 68% of Me-BPA has been released, with full conversion being achieved after 6 h.

## Mechanism of catalyst activation

In order to investigate the influence of phenols further, triphos-Ru-TMM was left to react with 4 equiv of the more acidic 2-fluorophenol. After 4 h at 60 °C, full conversion to a new ruthenium species with a $^{31}$P NMR signal at 49.9 ppm was noted. Single crystals of the deep red compound suitable for X-ray crystallographic analysis were obtained, revealing a triphos ruthenium diphenolate species **Ru-1** (Fig. 4a). This complex displays a distorted square pyramidal geometry in the crystal. As only a singlet phosphorous signal can be detected in the $^{31}$P NMR spectrum, the complex most likely displays a fluxional geometry in solution comparable to N-triphos-Ru(CO)$_2$[36]. Surprisingly, **Ru-1** reacted rapidly with THF-$d_8$ forming a ruthenium propionate (Supplementary Fig. 8). The analogous 2,5-difluorophenolate complex **Ru-2** was also prepared. Here, complete conversion was observed within 1 h of reaction time. As the acidity of the respective phenol appears to have a strong influence on the rate of formation of the Ru-diphenolates, the reaction is likely initiated by the protonation of the TMM ligand, forming a methyl allyl complex[25]. Ruthenium allyl can be amphiphilic, operating either as a nucleophile or electrophile[37]. As the more acidic 2,5-difluorophenol compared to Me-BPA is by far the fastest to complete this transformation, it seems reasonable that the 2-methylallyl ligand acts as a base and is protonated again. As such, the ruthenium diphenolate is generated under release of isobutene.

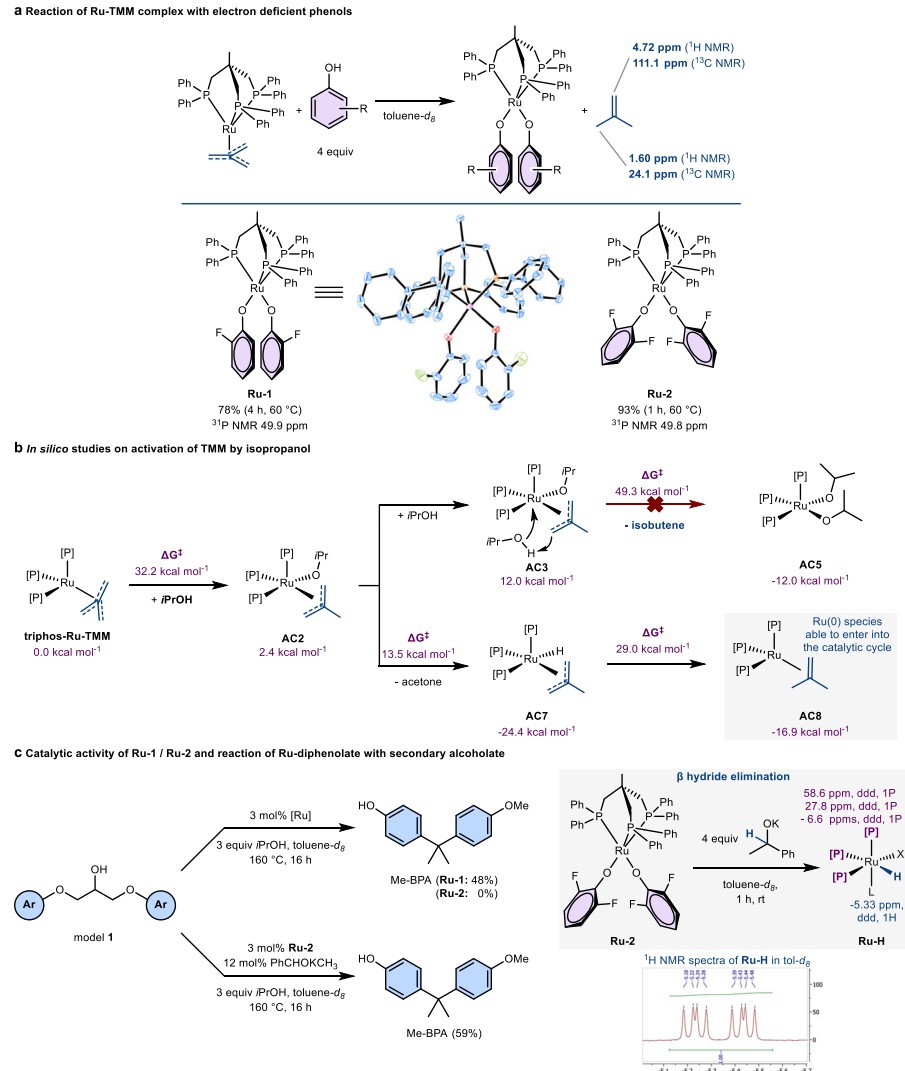

**Fig. 4 | Formation and reactivity of ruthenium diphenolates. a** Formation of diphenolate complex from triphos-Ru-TMM and electron deficient phenols. Molecular structure of **Ru-1** in the crystal (CCDC 2289877). **b** Activation mechanism of TMM precatalyst by phenol or isopropanol. The reported energies were calculated at the M06-D3/def2TZVP(SMD solvent = toluene)//M06-D3/def2SVP level of theory (see supplementary information, section 4.1 for detailed information). **c** Formation of hydride complex from diphenolate complex via β hydride elimination. Yields determined using ¹H NMR spectroscopy using 1,3,5-trimethoxybenzene as internal standard. ¹H NMR spectrum excerpt of **Ru-H** in toluene-$d_8$. d doublet, TMM trimethylenemethane, triphos 1,1,1-tris(diphenyl-phosphinomethyl)ethane, Me-BPA O-methyl bisphenol A.

Indeed, in operando experiments, the generation of isobutene was observed using ¹H and ¹³C NMR spectroscopy (Fig. 4a). Strong acids with weakly coordinating anions such as HNTf₂ selectively protonate the TMM ligand a single time[25] is due to the unfavoured formation of an unstablilised dicationic triphos-Ru complex upon loss of isobutene.

As the role of isopropanol in the induction period remained experimentally elusive, in silico studies were employed (Fig. 4b). A consecutive double protonation of TMM by isopropanol leading to a diisopropanolate complex **AC5**, which could not be observed spectroscopically, was found to have an impassable barrier of 49.3 kcal mol⁻¹ for the second protonation. However, the intermediary isopropanolate complex **AC2** formed from the first protonation can lead to acetone via a β-hydride elimination. For the resulting hydride species **AC7**, reductive elimination of isobutene was found, resulting in a Ru(0) species that plausibly could enter into the catalytic cycle for the dehydrogenation and C−O bond scission. The barrier for the reductive elimination is 29.0 kcal mol⁻¹, which, together with the similarly high barrier for the first protonation of 32.2 kcal mol⁻¹, aligns with the sluggish initiation phase of the catalysis. Interestingly, the protonation of the Ru-TMM fragments by model **S**, shown in Fig. 5, was

found to have a ΔG‡ of 38.5 kcal mol⁻¹ (Supplementary Fig. 13). This aligns well with the experimentally observed need for isopropanol to start the catalyst activation reaction (Fig. 2a). Thus, during the induction period of the catalysis, only small amounts of active species will be formed through isopropanol. However, once phenols are liberated, the alternative activation pathway through diphenolate complexes becomes possible, accelerating catalyst activation.

To explain the disparity between lignin and epoxy models, benzylic alcohols would need to be able to react with the TMM ligand more competently than model **1**. Indeed, reacting triphos-Ru-TMM with two equiv of benzyl alcohol in toluene-$d_8$ at 160 °C overnight led to the emergence of a new species corresponding to a ³¹P NMR signal at 27.9 ppm, besides traces of triphos-Ru-CO-H₂. The majority of benzyl alcohol was converted into benzyl aldehyde.

**Role of phenolate complexes in catalytic cycle**

The isolated diphenolate complexes were tested as precatalysts for C−O bond cleavage on model **1**. Complex **Ru-1** generated a 48% yield of Me-BPA after 16 h, while **Ru-2** was found to be inactive (Fig. 4c). Although electron withdrawing groups on the phenolate precatalyst

**a** Mechanism of accepterless dehydrogenation

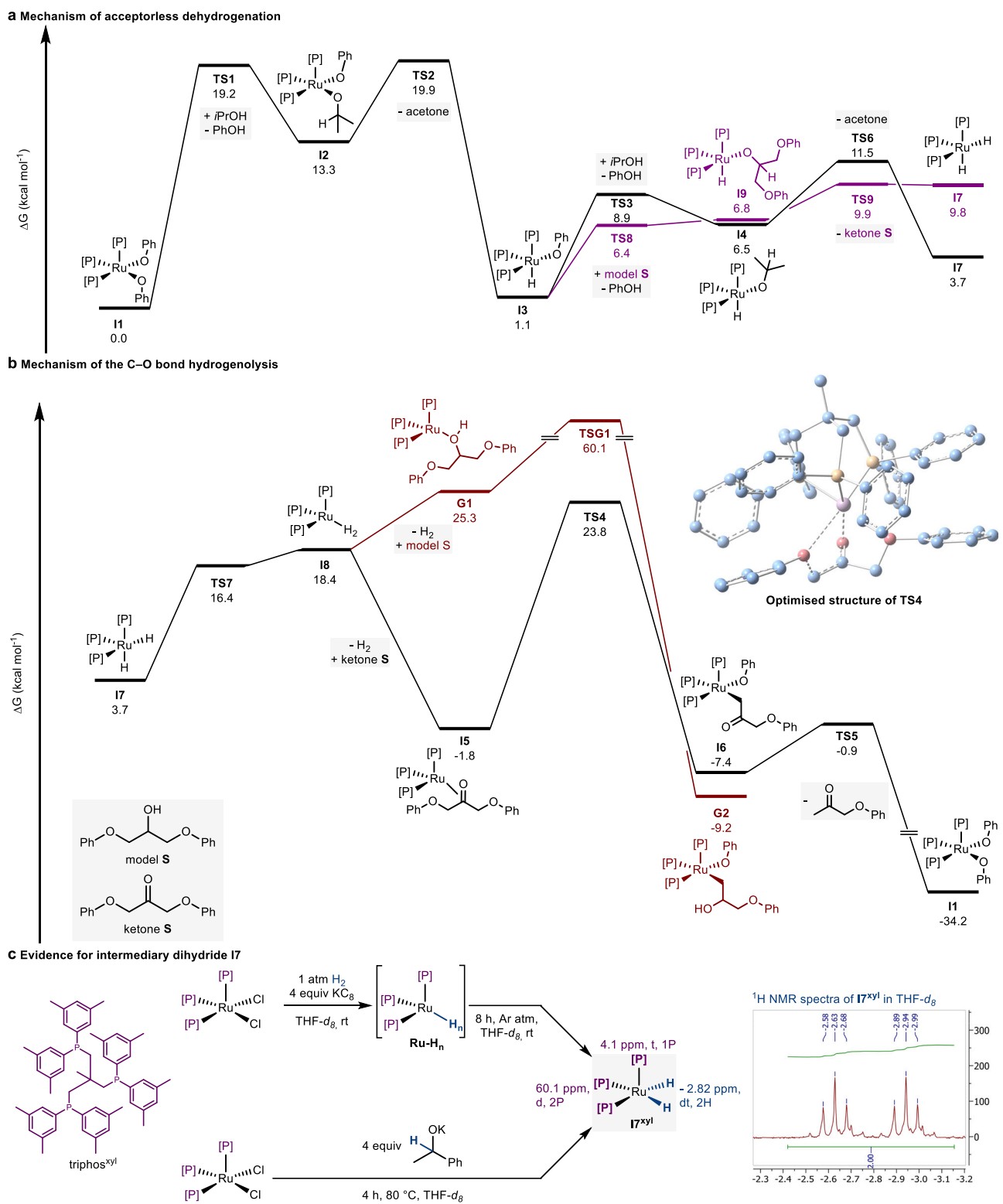

**b** Mechanism of the C–O bond hydrogenolysis

**c** Evidence for intermediary dihydride I7

**Fig. 5 | Gibbs free-energy profile of the catalytic cycle and experimental access to dihydride species. a** Dehydrogenation of model **S** or isopropanol starting from a Ruthenium phenolate. **b** Reductive C–O bond scission and catalyst regeneration. The reported energies were calculated at the M06-D3/def2TZVP(SMD solvent = toluene)//M06-D3/def2SVP level of theory (see supplementary information section 4.1 for detailed information). **c** Synthesis of **I7^xyl** via oxidative addition to dihydrogen and β hydride elimination. ¹H NMR spectrum excerpt of **I7^xyl** in THF-$d_8$. d doublet, t triplet, triphos^xyl (2-((bis(3,5-dimethylphenyl)phosphaneyl)methyl)-2-methylpropane-1,3-diyl)bis(bis(3,5-dimethylphenyl)phosphane).

reduce the yields, this corroborates the central role of the phenolates in the C–O bond scission. Reacting **Ru-1** with 20 equiv of isopropanol in toluene at 160 °C over the course of 4 h led to the formation of ~40% of acetone, as confirmed by ¹H NMR spectroscopy. Here, no new ruthenium species could be detected. It can be postulated that isopropanol coordinates ruthenium realising phenol. Thereafter, the alcoholate can form acetone via β-hydride elimination. That no isopropanolate complex could be detected hints at a dynamic exchange,

with the phenolate being thermodynamically favoured at room temperature. In order to switch from an equilibrium to a quantitative transformation, isopropanol was exchanged with potassium 1-phenylethanolate and reacted with **Ru-2**. After 1 h at rt, $^1$H and $^{31}$P NMR spectroscopy revealed the formation of a new ruthenium species (**Ru-H**) being either a square pyramidal or an octahedral mono hydride complex (Fig. 4c), as concluded by the ddd hydride peak at -5.33 ppm and the three corresponding phosphine signals ($^1$H, $^{31}$P HMBC NMR spectrum). Unlike in the case of using **Ru-2** alone, the Ru species **Ru-H** obtained from **Ru-2** allowed cleaving the C−O bonds in model **1**. This supports the entry of phenolate complexes into the catalytic cycle via β hydride elimination of isopropanol.

## Formulation of the catalytic cycle

Based on these experimental results, potential reaction mechanisms with defined Ru-species can be formulated. In silico studies were employed to gauge the free energy profiles of potential pathways. With the cost-effectiveness of the DFT calculations in mind, the methyl bisphenol A appendages were replaced by plain phenyl groups (Fig. 5a, more details in supplementary information, section 4.1). Investigations were started on the acceptorless dehydrogenation of simplified model **S** or isopropanol. The triphos-ruthenium diphenolate complex **I1** presents the start point into the catalytic cycle. Exchanging one of the phenolate ligands with an isopropanolate yielding **I2** increases the energy by 13.3 kcal mol$^{-1}$ with an activation barrier of 19.2 kcal mol$^{-1}$. This exchange can be expected to be dynamic and relatively fast at 160 °C, thus providing an explanation for why the corresponding isopropanolate complexes could not be detected in the previous in operando monitoring experiments at rt. **I2** can then extrude acetone under formation of the hydride species **I3** in an exergonic reaction by −12.2 kcal mol$^{-1}$. This species correlates to the structure of **Ru-H** detected by $^1$H and $^{31}$P NMR spectroscopy. From this complex, a second ligand exchange of phenolate for isopropanolate can take place (black). Alternatively, the alcohol found in the epoxy linkage can protonate the phenolate, forming **I9** (violet). In both cases, β-hydride elimination under extrusion of the corresponding ketone leads to the dihydride complex **I7**, with slightly different energies depending on the corresponding ketone released. The alcoholate based on the linkage motif **I9** is thermodynamically over the isopropanolate complex **I4**. However, the β-hydride elimination forming acetone is 2.8 kcal mol$^{-1}$ downhill, while the analogous formation of ketone **S** is 3 kcal mol$^{-1}$ uphill. The reductive elimination of phenol from **I3** was also considered as a potential pathway leading to a Ru(0) species; however, the activation barrier was found to be 35.6 kcal mol$^{-1}$ (Supplementary Fig. 20).

Subsequently, we continued the in silico studies on the C−O bond disconnection, which is central to the valorisation of lignin and the deconstruction of epoxy polymers. (Fig. 5b). Both dehydrogenation pathways leading to **I7** are viable. We chose to continue from 3.7 kcal mol$^{-1}$ as isopropanol is present in excess. It was found that the dihydride species **I7** can form a Ru(0) dihydrogen complex **I8**, which is energetically uphill by 14.7 kcal mol$^{-1}$. This electron rich species can then coordinate either model S or ketone S under the release of dihydrogen. The alcohol complex **G1** is higher in energy than the corresponding ketone complex **I5** by 27.1 kcal mol$^{-1}$. For both **G1** and **I5**, the oxidative addition of the Ru metal centre into the respective C−O bonds was found to be thermodynamically favoured. However, the transition state for the oxidative addition starting from the alcohol complex **G1** has an energy of 60.1 kcal mol$^{-1}$ above **I1**, which is unfeasible at 160 °C. On the other hand, the transition state for the corresponding oxidative addition into the σ-phenoxyketone was found at 23.8 kcal mol$^{-1}$ using the same energy reference. Both for the formation of **I5** and the oxidative addition through **TS4**, π-coordination of the ketone moiety by ruthenium(0) was found to stabilise the respective electron rich complexes. This can be explained by the η$^2$

ketone ligand allowing backdonation from the electron-rich Ru(0) complex.

Analysis of bond dissociation energies for model **1** and ketone **I** revealed that the homolytic C−O bond scission is 10.4 kcal mol$^{-1}$ lower for the ketone[11]. A similar difference has been observed for the β-O-4 linkage in lignin[38]. While this BDE difference has been invoked to rationalise the easier C−O bond activation adjacent to the carbonyl group of the ketone due to its reduced bond strength. The in silico analysis of the Ru-catalysed two-electron disconnection mechanism reveals that both the insertion of the Ru(0) complex into σ phenoxyalcohol model **S** and σ phenoxyketone ketone **S** is thermodynamically favoured. However, the energies of intermediates **I5** and **G1** show the large influence of ketone **S** or model **S** on stabilising the Ru(0) species. Furthermore, the energy barrier for the C−O bond cleavage is 9.2 kcal mol$^{-1}$ lower with the ketone (25.6 kcal mol$^{-1}$ compared to 60.1 kcal mol$^{-1}$ with the alcohol). Therefore, rather than C−O bond strength, the selectivity is controlled by the coordination of the substrate to the Ru(0) species and the barrier of oxidative addition. After the oxidative addition, the Ru−C bond of complex **I6** can be protonated by a phenol, regenerating **I1** and releasing the first cleavage product. The regeneration of the starting complex is strongly downhill over a considerably small barrier of 6.5 kcal mol$^{-1}$.

We attempted to synthesise the intermediary dihydride species **I7** to corroborate its role in the catalysis. In order to avoid the formation of hydride bridged ruthenium species, a more sterically hindered triphos variation designed by Leitner, Klankermeyer et al.[23] was used for these experiments. triphos$^{xyl}$-Ru-Cl$_2$ was chosen as the template complex for accessing **I7$^{xyl}$**. Reducing the dichloro complex with KC$_8$ under 1 atm of dihydrogen[39] resulted in the formation of an intermediary hydride species **Ru-H$_n$**, which most likely corresponds to a ruthenium dihydride/dihydrogen or tetrahydride species[40]. The generation of an intermediary Ru(0) species that oxidatively adds presents the microscopic reversion of the formation of **I8** through **TS7**. At room temperature under argon atmosphere, this complex slowly converts to the dihydride species **I7$^{xyl}$**, as confirmed by $^1$H and $^{31}$P NMR spectroscopy (Fig. 5c). In the catalytic C−O bond scission mechanism, **I7** forms from subsequent β-hydride elimination of secondary alcohols. Indeed, reacting triphos$^{xyl}$-Ru-Cl$_2$ with potassium 1-phenylethan-1-olate also lead to the formation of **I7$^{xyl}$** analogously to the calculated mechanism (Fig. 5c). The observed reactivity aligns perfectly with the catalytic cycle.

## Probing C−N bond scission

Amine-cured epoxy resins are widely used and, therefore, important targets for chemical deconstruction[11]. The curing process with alkyl amines results in C−N bonds adjacent to the central alcohol motif. As these bonds are structurally analogous to the here discussed C−O bonds, alkyl amine model **2** was designed in order to probe the chemoselectivity of the catalytic system (Fig. 6). Besides Me-BPA, two amines could be isolated together and identified. Amine **1** corresponds to the reduced ketone that is initially cleaved off and furthermore reductively alkylated with acetone. Amine **2**, on the other hand, forms from the cleaved of ketone via reductive amination of itself.

No evidence of C−N bond scission was observed. Although a considerable amount of amine products could not be recovered, this is most likely due to the formation of higher oligomers from the ketone intermediate.

## Discussion

In summary, we report a comprehensive mechanism for the ruthenium catalysed C−O bond hydrogenolysis using triphos-Ru-TMM precatalysts, which is relevant to lignin valorisation and epoxy resin deconstruction. Both for catalyst activation and the catalytic cycle, unreported phenolate complexes play a central role, meaning that the reaction product itself is key to forming the active species.

**Fig. 6 | Deconstruction of model 2.** Investigation of potential C−N bond scission by triphos-Ru-TMM. TMM trimethylenemethane, triphos 1,1,1-tris(diphenyl-phosphi-nomethyl)ethane, Me-BPA O-methyl bisphenol A.

Furthermore, theoretical and experimental investigations revealed the formation of a triphos-Ru(0) dihydrogen complex from a dihydride complex, which then activates the targeted C−O bond via an oxidative addition. The here reported studies may allow for improving the reported methods for lignin valorisation and epoxy resin deconstruction. Furthermore, the design of other Ru-catalysed transformations might be aided by the revealed mechanistic insights of this work.

## Methods

Unless otherwise stated, all reactions were set up and worked up in a glovebox under an atmosphere of argon. All chemicals were purchased from Sigma-Aldrich, Tokyo Chemical Industry (TCI) or Strem Chemicals and used as received. Tetrakis(dimethylsulfoxide)dichlororuthenium(II) was donated by Heraeus Precious Metals GmbH & Co. KG and used as received. THF, toluene, $CH_2Cl_2$ and MeCN were retrieved from a MBraun SP-800 purification system, degassed using argon and stored over 3 Å molecular sieves. The remaining solvents were purchased from Sigma-Aldrich degassed using argon, stored over 3 Å molecular sieves and used without further purification.

### Reaction set up

Operando monitoring experiments using NMR spectroscopy were set up in an Argon-charged glovebox using Low Pressure Vacuum NMR tubes (5 mm O.D., 177,8 mm) purchased from SP Wilmad-LabGlass. Reactions were heated in a silicone oil bath. Preparative / deconstruction reactions were set up in an Argon-charged glovebox using a 10 ml COtube (pressure tube approved up to 5 bar) sealed with PTFE/silicon seals purchased from SyTracks as reaction vessel and using a Teflon-coated stirring bar. Reactions were stirred in metal heating blocks at 650 rpm.

### Nuclear Magnetic Resonance spectroscopy

1H NMR, 13C NMR, 19F NMR and 31P NMR spectra were recorded on a Bruker 400 MHz Ascend spectrometer. Chemical shifts were given as δ value (ppm) with reference to residual solvent signal of the deuterated solvent. The peak patterns are indicated as follows: s, singlet; d, doublet; t, triplet; m, multiplet; q, quartet. Multiplicities reported for 13C NMR spectra were assigned using APT, DEPT-90 and DEPT-135 spectra. Chemical structures were identified with the help of 1H, 1H COSY; 1H,13C HSQC; 1H,13C HMBC and 1H,31P HMBC NMR experiments. The coupling constants, J, are reported in Hertz (Hz). The spectra were calibrated to the residual solvent signals1. NMR spectra were processed with MestReNova Version 14.2.1−27684.

### Representative synthesis of model compounds (model 2)

In a round bottom flask under air, 307 mg (1.03 mmol, 1 equiv) of 2-((4-(2-(4-methoxyphenyl)propan-2-yl)phenoxy)methyl)oxirane were mixed with 1.30 ml (1.25 g, 10.3 mmol, 10 equiv) of 2-phenylethan-1-amine and stirred at 65 °C over-night, then cooled to room temperature. The residue was taken up in 20 ml of ethyl acetate and washed five times with 10 ml of brine, dried over $MgSO_4$ and filtered. The solvent volume was reduced to a few ml in vacuo, overlayered with pentane and then

stored in a freezer at -30 °C. The product precipitated as colourless solid over the course of a few days. The mother liquor was decanted off and the solid washed with pentane, affording the product in a yield of 65% (282 mg, 0.67 mmol).

1H NMR (400 MHz, DCM-$d_2$): δ 7.31 − 7.26 (m, 2H), 7.23 − 7.17 (m, 3H), 7.14 − 7.10 (m, 4H), 6.82 − 6.75 (m, 4H), 3.97 (dtd, J = 10.8, 5.8, 3.0 Hz, 1H), 3.93 − 3.89 (m, 2H), 3.76 (s, 3H), 3.00 − 2.69 (m, 6H), 1.62 (s, 6H).; 13 C NMR (CDCl₃, 101 MHz, 25 °C): δ = 157.9 (s, 1 C), 157.0 (s, 1 C), 143.9 (s, 1 C), 143.5 (s, 1 C), 140.4 (s, 1 C), 129.1 (d, 2 C), 128.8 (d, 2 C), 128.1 (d, 2 C), 128.0 (d, 2 C), 126.5 (d, 1 C), 114.2 (d, 2 C), 113.5 (d, 2 C), 70.9 (t, 1 C), 68.4 (d, 1 C), 55.5 (q, 1 C), 51.9 (t, 1 C), 51.3 (t, 1 C), 41.9 (s, 1 C), 36.6 (t, 1 C), 31.1 (q, 2 C) ppm; HRMS (ESI⁺): calculated [M + H]⁺ = [$C_{27}H_{33}NO_3$ + H]⁺ 420.2533; found 420.2611.

### Representative procedure for catalytic deconstruction reactions

43.3 mg (0.08 mmol, 1 equiv) of model **1** or 23.9 mg (0.08 mmol, 1 equiv) of ketone **III** were dissolved in 0.2 ml of toluene-$d_8$ and 18.3 μl (14.4 mg, 0.24 mmol, 3 equiv) of isopropanol, then 3 mol% (2.40 μmol) catalyst were added. After sealing the reaction vessel, the mixtures were stirred outside of the glovebox in an aluminium heating block at 160 °C. After the given reaction time (16 h as standard reaction time), 1,3,5-trimethoxybenzene was added to the reaction mixture under argon. Yields were determined by 1H NMR and 31P NMR spectroscopy of the crude mixture with 1,3,5-trimethoxybenzene as internal standard in toluene-$d_8$.

**Compounds monitored in reaction mixture.** Model **1** 1H NMR (toluene-$d_8$, 400 MHz, 25 °C): δ = 7.10 − 7.06 (m, 8H), 6.77 − 6.67 (m, 9H), 4.01 − 3.96 (m, 1H), 3.58 − 3.46 (m, 4H), 3.36 (s, 6H), 1.56 (s, 12H) ppm. ketone **III** 1H NMR (toluene-$d_8$, 400 MHz, 25 °C): δ = 7.07 − 7.01 (m, 4H), 6.72 − 6.70 (m, 2H), 6.61 − 6.59 (m, 2H), 3.96 (s, 2H), 1.80 (s, 3H), 1.55 (s, 6H) ppm. alcohol **III** 1H NMR (toluene-$d_8$, 400 MHz, 25 °C): δ = 7.11 − 7.07 (m, 4H), 6.74 − 6.68 (m, 4H), 3.96 − 3.91 (m, 1H), 3.58 − 3.49 (m, 2H), 1.59 (s, 6H), 1.09 (d, J = 6.4 Hz, 3H) ppm. Me-BPA 1H NMR (toluene-$d_8$, 400 MHz, 25 °C): δ = 7.11 − 7.03 (m, 4H), 6.85 − 6.78 (m, 2H), 6.72 − 6.65 (m, 2H), 3.37 (s, 3H), 1.55 (s, 6H) ppm.

### For measuring kinetic profiles

Model **1** was reacted nine times with different reaction times each. 43.3 mg (80.0 μmol, 1 equiv, as stock solution) of model **1** and 0.2 ml of toluene-$d_8$ were given into a 10 ml COtube in an Argon-charged glovebox. Potential additives were also added to that stock solution. 3 mol % (1.87 mg, 2.40 μmol) of triphos-Ru-TMM and 3 equiv (14.4 mg, 18.3 μl, 240 μmol) of isopropanol were added. After sealing the reaction vessel, the mixtures were stirred at 650 rpm outside of the glovebox in aluminium heating blocks at 160 °C. After the given reaction time, the reaction was cooled using a water/ice bath. Then, 1,3,5-trimethoxybenzene (as stock solution in toluene-$d_8$) was added to the reaction mixture. Yields were determined by 1H NMR spectroscopy of the reaction mixture with 1,3,5-trimethoxybenzene as internal standard. GC-MS was used to confirm the products detected via 1H NMR spectroscopy for all entries.

## Data availability

The materials and methods, experimental procedures, mechanistic studies, characterisation data, and NMR spectra data generated in this study are provided in the Supplementary Information file. The calculated coordinate data generated in this study are provided in the source data file. The X-ray crystallographic coordinates for structures reported in this study have been deposited at the Cambridge Crystallographic Data Centre (CCDC), under deposition numbers 2289877, 2303540, and 2289791. These data can be obtained free of charge from The Cambridge Crystallographic Data Centre via www.ccdc.cam.ac.uk/data_request/cif. All data is available from the corresponding authors. The authors declare that all other data supporting the findings of this study are available within the paper and its Supplementary Information files. Source data are provided with this paper.

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

## Acknowledgements

We are deeply grateful for the financial support by Innovation Fund Denmark (Grant 0224-00072B, T.S.), Danish National Research Foundation (Grant No. DNRF118 and DNRF93, T.S.), the Novo Nordisk Foundation (grant no. NNF21SA0072700, T.S.) and Aarhus University. Support from the European Union's Horizon 2020 research and innovation programme under grant agreement No 862179 (T.S.) and Marie Sklodowska-Curie grant agreement No 859910 (T.S.) is also gratefully acknowledged. This publication reflects the views only of the authors, and the Commission cannot be held responsible for any use that may be made of the information contained therein. Furthermore, we are grateful to the Research Council of Norway through the Centre of Excellence (No. 262695, A.N.) and FRIPRO project (No. 314321, A.N.) and the Norwegian Metacenter for Computational Science (NOTUR) for computational resources (project number nn4654k, A.N.). We are deeply grateful to Heraeus Precious Metals GmbH & Co. KG, who supported this study by providing ruthenium precursors. We thank CSCAA for the computing hours provided for the DFT study.

## Author contributions

Conceptualisation and writing/revising of the original draft were carried out by A.A. and T.S. Experiment design and investigation was carried out by A.A. and E.V.S. X-Ray crystallographic investigations were conducted by HCDH. Density Functional Theory studies were conducted by GMFB and supervised by A.N. Funding acquisition was carried out by T.S. All authors reviewed the final manuscript.

## Competing interests

T.S. is co-owner of SyTracks A/S, which commercialises the COtubes. The remaining authors declare no competing interests.
