## [Peer Review File · Nature Communications]

Unveiling the mechanism of triphos Ru catalysed C–O bond disconnections in polymersEditorial Note: This manuscript has been previously reviewed at another journal that is not operating a transparent peer review scheme. This document only contains reviewer comments and rebuttal letters for versions considered at Nature Communications.

Reviewers' Comments:

Reviewer #1:

Remarks to the Author:

[Note from the Editor: Reviewer #1 was asked to look over the response given to reviewer #3 who was not able to see the revision again.]

The authors have done a good job responding to the reviewer comments, and I think for the most part the technical concerns of the reviewers have been addressed satisfactorily. On the point of the DFT level of theory, I agree with the authors that def2SVP is a generally sufficient basis set for geometry optimizations and frequency calculations, with single-point electronic energies calculated using the def2TZVP basis set. Optimization of all structures with M06-D3/def2TZVP would indeed be very computationally costly, and it is common to do what the authors have done. Still, I suggest following Reviewer 3's request to recalculate with def2TZVP the geometries (and free-energy corrections) of the few intermediates and transition states that define the overall free-energy barrier for catalysis.

I also think it is worth looking at solvent-coordinated variants of the 5-coordinate complexes as suggested by Reviewer 3. The authors argue convincingly that these complexes are sterically crowded already, but it is straightforward to test whether a bound solvent might lower the free energy of an intermediate, which might change the overall energy barrier.

Separate is the question of whether the impact and broad interest of this work are in keeping with the standards of Nature Communications. I will admit that I don't have a clear sense for how this journal compares with others that I'm more familiar with, so I hesitate to make a definitive call on this. As I am more familiar with the ACS family of journals, I will say that this paper would be right at home in Organometallics, and would likely have trouble getting into ACS Catalysis.

Minor comment:

The yield of Me-BPA in Figure 2a now says "<99%". Is it meant to say ">99%"?

Reviewer #2:

Remarks to the Author:

The authors present a combined experimental, spectroscopical and theoretical studies on triphos-Ru-TMM catalyzed C–O bonds hydrogenolysis in lignin models and epoxy polymers. The revised version has been greatly improved, and well revealed the details of the catalytic cycle on molecular level. Theoretical and experimental investigations revealed that triphos ruthenium phenolate complexes play a central role in both catalyst activation and the catalytic cycle. In addition, the generation and identification of the catalytic active species has been well elucidated with computation and X-ray crystals. This mechanistic study may be of great significance for the future design of novel Ru-catalyzed transformations. In general, I believe that this version is suitable for publication in Nature Communications.

Please find our point-by-point reply to the reviewer's comments below. Our answers to the comments are written in blue.

.....

Reviewer #1:

The authors have done a good job responding to the reviewer comments, and I think for the most part the technical concerns of the reviewers have been addressed satisfactorily. On the point of the DFT level of theory, I agree with the authors that def2SVP is a generally sufficient basis set for geometry optimizations and frequency calculations, with single-point electronic energies calculated using the def2TZVP basis set. Optimization of all structures with M06-D3/def2TZVP would indeed be very computationally costly, and it is common to do what the authors have done. Still, I suggest following Reviewer 3's request to recalculate with def2TZVP the geometries (and free-energy corrections) of the few intermediates and transition states that define the overall free-energy barrier for catalysis.

Our answer: As suggested by the reviewer, we evaluated the influence of the basis set in our energies further. We focused on the H₂ reductive elimination. This step was selected because the structures involved (I7, TS7, and I8) are the smallest in size (86 atoms) compared to other key transition states (TS4, 116 atoms), and just the optimization and frequency steps already took approximately 1 week to be accomplished. The structures were optimized at the def2TZVP level of theory, and energy refinements were done at the def2TZVP and def2QZVP levels. Small energy differences of 1 kcal/mol for TS7 and 0.3 kcal/mol for I8 were obtained by using def2TZVP or def2SVP for the optimization, while for the energy refinement, the use of a def2QZVP leads to no benefit. We think these results strongly corroborate that the accuracy and computational cost for the selected methodology are well balanced. This data was added in the computational section in the revised supplementary information:

“The energy barrier from I7 to I8 was selected for comparing results using different basis sets (Fig. S13). Our results show that def2-QZVP for energy refinement returns no changes in the energy barriers. Additionally, using a triple-zeta def2-TZVP basis set for geometry optimization led to a small change in the barrier of 1 kcal/mol, albeit at a much higher computational cost. These results indicate that the accuracy and computational cost for the selected methodology are well balanced.

Fig. S13. To the right is the comparison of the ΔG^\ddagger for TS7 at different computational levels.”

I also think it is worth looking at solvent-coordinated variants of the 5-coordinate complexes as suggested by Reviewer 3. The authors argue convincingly that these complexes are sterically crowded already, but it is straightforward to test whether a bound solvent might lower the free energy of an intermediate, which might change the overall energy barrier.

Our answer: As suggested by the reviewer, we computationally probed the stability of the 5-coordinate complexes by adding other ligands (L = iPrOH, Acetone, PhOH, Model-S, Ketone-S) to Ru-S (I1 in the revised manuscript) and I3. However, in all cases and as predicted, the hexacoordinated species was less stable than the free molecules (e. g.: Ru-S (I1) + L), likely due to the previously discussed steric crowdedness at the coordination centre. A comment about this was added in the computational section in the supporting information:

“Furthermore, we analyzed the possibility of organic molecules present in the medium to coordinate with intermediates, which could lead to lower energy complexes (Scheme. S1). This analysis was performed using I1, I3, and the coordinating molecules present in the system (iPrOH, PhOH, Model S, Ketone S, Acetone). Despite the coordination ability of these organic molecules, the higher-order coordination complexes were less stable than the separate molecules. Analysis of I7 with Ketone S or acetone led to the same conclusion. The same was observed in the complex of I4 with phenol.

1) Example of calculation using I1 and PhOH

2) Data for different intermediates

2	ΔG_f
I1 + iPrOH	16.2
I1 + PhOH	3.5
I1 + Acetone	14.9
I1 + ModelS	10.4
I1 + KetoneS	12.7
I3 + iPrOH	2.4
I3 + PhOH	3.7
I3 + Acetone	3.5
I3 + ModelS	2.5
I3 + KetoneS	3.0
I7 + KetoneS	1.8
I7 + Acetone	7.1
I4 + PhOH	2.6

Scheme S1. Computed ΔG for the formation complexes Ru-S or I3 with one additional ligand; iPrOH, PhOH, Model S, Ketone S, or Acetone.“

Separate is the question of whether the impact and broad interest of this work are in keeping with the standards of Nature Communications. I will admit that I don't have a clear sense for how this journal compares with others that I'm more familiar with, so I hesitate to make a definitive call on this. As I am more familiar with the ACS family of journals, I will say that this paper would be right at home in Organometallics, and would likely have trouble getting into ACS Catalysis.

Minor comment:

The yield of Me-BPA in Figure 2a now says "<99%". Is it meant to say ">99%"?

Our answer: This error was corrected in the revised manuscript.

Reviewer

#2:

The authors present a combined experimental, spectroscopical and theoretical studies on triphos-Ru-TMM catalyzed C–O bonds hydrogenolysis in lignin models and epoxy polymers. The revised version has been greatly improved, and well revealed the details of the catalytic cycle on molecular level. Theoretical and experimental investigations revealed that triphos ruthenium phenolate complexes play a central role in both catalyst activation and the catalytic cycle. In addition, the generation and identification of the catalytic active species has been well elucidated with computation and X-ray crystals. This mechanistic study may be of great significance for the future design of novel Ru-catalyzed transformations. In general, I believe that this version is suitable for publication in Nature Communications.

+++